# Antitumor Activities of a Humanized Cancer-Specific Anti-HER2 Monoclonal Antibody, humH_2_Mab-250 in Human Breast Cancer Xenografts

**DOI:** 10.3390/ijms26031079

**Published:** 2025-01-26

**Authors:** Mika K. Kaneko, Hiroyuki Suzuki, Tomokazu Ohishi, Takuro Nakamura, Miyuki Yanaka, Tomohiro Tanaka, Yukinari Kato

**Affiliations:** 1Department of Antibody Drug Development, Tohoku University Graduate School of Medicine, 2-1 Seiryo-machi, Aoba-ku, Sendai 980-8575, Miyagi, Japan; mika.kaneko.d4@tohoku.ac.jp (M.K.K.); takuro.nakamura.a2@tohoku.ac.jp (T.N.); miyuki.yanaka.c5@tohoku.ac.jp (M.Y.); tomohiro.tanaka.b5@tohoku.ac.jp (T.T.); 2Institute of Microbial Chemistry (BIKAKEN), Numazu, Microbial Chemistry Research Foundation, 18-24 Miyamoto, Numazu 410-0301, Shizuoka, Japan; ohishit@bikaken.or.jp; 3Institute of Microbial Chemistry (BIKAKEN), Laboratory of Oncology, Microbial Chemistry Research Foundation, 3-14-23 Kamiosaki, Shinagawa-ku 141-0021, Tokyo, Japan

**Keywords:** cancer-specific monoclonal antibody, HER2, ADCC, CDC, xenograft, breast cancer

## Abstract

Monoclonal antibody (mAb) and cell-based immunotherapies represent cutting-edge strategies for cancer treatment. However, safety concerns persist due to the potential targeting of normal cells that express reactive antigens. Therefore, it is crucial to develop cancer-specific mAbs (CasMabs) that can bind to cancer-specific antigens and exhibit antitumor activity in vivo, thereby reducing the risk of adverse effects. We previously screened mAbs targeting human epidermal growth factor receptor 2 (HER2) and successfully developed a cancer-specific anti-HER2 mAb, H_2_Mab-250/H_2_CasMab-2 (mouse IgG_1_, kappa). In this study, we assessed both the in vitro and in vivo antitumor efficacy of the humanized H_2_Mab-250 (humH_2_Mab-250). Although humH_2_Mab-250 showed lower reactivity to HER2-overexpressed Chinese hamster ovary-K1 (CHO/HER2) and breast cancer cell lines (BT-474 and SK-BR-3) than trastuzumab in flow cytometry, both humH_2_Mab-250 and trastuzumab showed similar antibody-dependent cellular cytotoxicity (ADCC) against CHO/HER2 and the breast cancer cell lines in the presence of effector splenocytes. In addition, humH_2_Mab-250 exhibited significant complement-dependent cellular cytotoxicity (CDC) in CHO/HER2 and the breast cancer cell lines compared to trastuzumab. Furthermore, humH_2_Mab-250 possesses compatible in vivo antitumor effects against CHO/HER2 and breast cancer xenografts with trastuzumab. These findings highlight the distinct roles of ADCC and CDC in the antitumor effects of humH_2_Mab-250 and trastuzumab and suggest a potential direction for the clinical development of humH_2_Mab-250 for HER2-positive tumors.

## 1. Introduction

Monoclonal antibody (mAb)-based therapeutics are essential for treating various diseases. The U.S. Food and Drug Administration (FDA) approved the first therapeutic mAb, Orthoclone OKT3 (mouse anti-CD3 mAb), for kidney transplantation rejection in 1986 [1]. However, the first-generation mouse mAbs tested in clinical trials had limited effectiveness due to their immunogenicity and poor effector functions [2]. Patients developed human anti-mouse antibody responses, which caused the rapid clearance of the therapeutic mAbs from the body and restricted the number of possible treatment doses [2]. The creation of engineered chimeric, humanized, and fully human mAbs has uncovered several valuable applications for antibody-based therapies [3,4].

In mAb therapy for solid tumors, the FDA approved trastuzumab for human epidermal growth factor receptor 2 (HER2)-positive breast cancer in 1998 [5]. HER2-positive breast cancer is defined by circumferential membrane staining that is complete, intense, and in >10% of tumor cells in immunohistochemistry (IHC 3+) and/or in situ hybridization (ISH)-positive [6]. Trastuzumab is a humanized mAb by inserting the complementarity determining regions (CDRs) of mouse anti-HER2 mAb (clone 4D5) into the framework of a consensus human IgG_1_ [7]. Trastuzumab exhibited antitumor efficacy against HER2-positive breast cancer xenograft in monotherapy or combination therapy with chemotherapy [8,9,10]. The clinical efficacy of trastuzumab is mediated by the immunologic engagement [11]. Trastuzumab exerts antibody-dependent cellular cytotoxicity (ADCC) upon the binding to Fcγ receptors on natural killer cells or macrophages [11]. The combination therapy of trastuzumab with chemotherapy improves the progression-free survival and overall survival in HER2-positive breast cancer patients with metastasis [12]. Currently, HER2 overexpression and activating mutations have been observed in gastric and gastroesophageal cancers [13,14], endometrial cancers [15,16], non-small-cell lung cancers [17,18], and ovarian cancers [19].

Trastuzumab-deruxtecan (T-DXd), a trastuzumab-based antibody–drug conjugate (ADC), has been developed and received the FDA approval [20]. T-DXd has demonstrated superior efficacy not only in HER2-positive breast cancers [21,22] but also in HER2-low (IHC 1+ or IHC 2+/ISH-non-positive) advanced breast cancers [23] and HER2-mutant non-small-cell lung cancers [24]. Given that approximately half of all breast cancers are classified as HER2-low, a substantial number of patients are expected to benefit from T-DXd therapy [25]. Although T-DXd is generally well-tolerated and rarely causes severe toxicity, studies have consistently linked it to the development of cardiac toxicity. While this issue is not clinically significant in most cases, baseline cardiac evaluation, regular monitoring, and early detection of cardiac adverse events are still crucial for T-DXd. Since HER2 plays a critical role in normal heart development and homeostasis [26,27], on-target, off-tumor toxicity in the heart would cause adverse effects. Therefore, management of the specificity of mAb to tumors will be required for further optimization.

Although 97 ADCs have been evaluated in clinical trials since 2000, 81 trials were terminated due to the lack of efficacy (32 agents) and safety issues (32 agents) [28]. On-target, off-tumor toxicity is thought to be a cause of adverse effects when the target antigen is expressed in normal cells. Therefore, selecting mAbs that specifically recognize cancer-related epitopes is critical to reducing unwanted side effects.

Activation of the complement-dependent cytotoxicity (CDC) pathway has been suggested as a mechanism to enhance the therapeutic effectiveness of antitumor mAbs [29]. It is one of the reported mechanisms for B-cell targeting anti-CD20 mAbs, such as ofatumumab and rituximab [30,31,32,33]. Trastuzumab mediates antitumor effects through various mechanisms but is unable to induce CDC in HER2-positive cells in the presence of human serum [34,35]. The activation of the classical complement pathway is regulated by various factors, including the size and density of the antigen, which influence the geometry of the antigen-antibody complex needed for effective C1q binding [29]. For optimal CDC activity, the Fc domains of antibodies within antigen-antibody clusters must be organized in a hexameric structure, which creates a geometry that enhances C1q binding and complement activation [29]. Approaches to improve CDC, such as antibody hexamerization [36,37] and Fc mutations [38], have shown promise in boosting antitumor activity in preclinical studies. For example, hexamerization was used to develop an anti-CD37 biparatopic antibody with enhanced in vitro CDC activity [39].

We previously generated cancer-specific anti-HER2 mAbs, H_2_Mab-214/H_2_CasMab-1 [40] and H_2_Mab-250/H_2_CasMab-2 [41], selected from 278 anti-HER2 clones, using HER2 expressed by glioblastoma LN229 cells as the target antigen. Interestingly, both H_2_Mab-214 and H_2_Mab-250 showed no reactivity toward spontaneously immortalized normal epithelial cells, such as HaCaT and MCF 10A [40,41]. Moreover, H_2_Mab-250 exhibited no binding to normal epithelial cells derived from various tissues, including the mammary gland, kidney proximal tubule, gingiva, colon, thymus, cornea, and lung bronchus [41]. In contrast, most anti-HER2 mAbs, including trastuzumab, reacted with both cancer and normal epithelial cells [41]. Furthermore, H_2_Mab-250 exhibited no reactivity with the normal heart in IHC [41]. Epitope mapping identified Trp614 in HER2 extracellular domain 4 (ECD4) as a critical determinant for H_2_Mab-250 recognition [41]. H_2_Mab-214 was also found to target a similar epitope as H_2_Mab-250, with structural analysis suggesting that H_2_Mab-214 binds to a misfolded region of the β-sheet in HER2-ECD4 [32]. This suggests that localized misfolding within the cysteine-rich portion of ECD4 contributes to the cancer specificity of H_2_Mab-214. Additionally, we engineered mouse IgG_2a_ and mouse-human chimeric versions of H_2_Mab-250. Both antibodies demonstrated antitumor activity against breast cancer xenografts in vivo, performing comparably to trastuzumab despite lower binding affinity and effector function activation in vitro [42,43].

This study evaluates the ADCC, CDC, and antitumor efficacy of the humanized version of H_2_Mab-250 (humH_2_Mab-250).

## 2. Results

### 2.1. Humanized Anti-HER2 mAb (humH_2_Mab-250)

We previously established an anti-HER2 mAb (H_2_Mab-250; mouse IgG_1_, kappa) by immunization with the HER2 ectodomain produced by glioblastoma LN229 cells [41]. H_2_Mab-250 was shown to be useful for flow cytometry [41]. In this study, we engineered a humanized H_2_Mab-250 (humH_2_Mab-250) by fusing the V_H_ and V_L_ CDRs of H_2_Mab-250 with the C_H_ and C_L_ chains of human IgG_1_, respectively (Figure 1A). We also engineered trastuzumab and humCvMab-62 (a control human IgG_1_) from CvMab-62 (mouse IgG_1_, an anti-SARS-CoV-2 spike protein S2 subunit mAb) [44]. Recombinant mAbs were produced by fucosyltransferase 8-knockout ExpiCHO-S cells to produce the defucosylated form of mAbs. The defucosylation of mAbs has been reported to potentiate the binding to the FcγRIIIa receptor, which results in increased ADCC activity by natural killer (NK) cells or macrophages [45,46]. In reduced conditions, we confirmed the purity of mAbs by SDS-PAGE (Figure 1B).

As shown in Figure 2, humH_2_Mab-250 and trastuzumab reacted with CHO/HER2 (HER2-overexpressed CHO-K1) cells in a dose-dependent manner (Figure 2A) but not with parental CHO-K1 cells (Figure 2B). Furthermore, humH_2_Mab-250 and trastuzumab reacted with HER2-positive breast cancer BT-474 (Figure 2C) and SK-BR-3 (Figure 2D). The reactivity of humH_2_Mab-250 to HER2-positive cells was similar compared to that of parental mAb, H_2_Mab-250 [41]. We also confirmed no reactivity of humH_2_Mab-250 to HaCaT (human keratinocyte), 293FT (embryonic kidney), and MCF 10A (mammary gland epithelial cells) (Appendix A) as we previously showed that using H_2_Mab-250 [41]. The humCvMab-62 did not react with CHO-K1, CHO/HER2, BT-474, and SK-BR-3 at 10 µg/mL (Appendix A). We used humCvMab-62 as a control human IgG_1_.

### 2.2. ADCC and CDC by humH_2_Mab-250 Against HER2-Positive Cells

We next examined whether humH_2_Mab-250 exerted ADCC activity against CHO/HER2 cells. Since human IgG_1_ can bind to all activating mouse Fcγ receptors and induce ADCC in the presence of mouse NK cells and macrophages [47], we evaluated ADCC of humH_2_Mab-250 and trastuzumab in the presence of mouse splenocytes as effector cells. As shown in Figure 3A, humH_2_Mab-250 and trastuzumab induced ADCC in the presence of effector splenocytes against CHO/HER2 (19.9 and 23.0% cytotoxicity, respectively) more effectively than the control human IgG_1_ (5.9% cytotoxicity; *p* < 0.05). Furthermore, both humH_2_Mab-250 and trastuzumab induced ADCC against BT-474 (8.8 and 9.9% cytotoxicity, respectively) more effectively than the control human IgG_1_ (1.7% cytotoxicity; *p* < 0.05, Figure 3B). Both humH_2_Mab-250 and trastuzumab also induced ADCC against SK-BR-3 (5.5 and 5.8% cytotoxicity, respectively) more effectively than the control human IgG_1_ (1.4% cytotoxicity; *p* < 0.05, Figure 3C). We also examined the ADCC activity against CHO/HER2, BT-474, and SK-BR-3 cells in the presence of human NK cells. As shown in Appendix A, both humH_2_Mab-250 and trastuzumab induced significant ADCC against CHO/HER2. Additionally, trastuzumab induced significant ADCC against BT-474, and humH_2_Mab-250 induced significant ADCC against SK-BR-3. These results suggest that humH_2_Mab-250 exerted compatible ADCC activities against HER2-positive cells by mouse splenocytes compared to that by human NK cells.

We investigated CDC by humH_2_Mab-250 and trastuzumab against CHO/HER2. As shown in Figure 4A, humH_2_Mab-250 showed a significant CDC in the presence of complements against CHO/HER2 (14.4% cytotoxicity) more effectively than the control human IgG_1_ (3.5% cytotoxicity; *p* < 0.05). In contrast, trastuzumab did not show a significant difference compared to the control in CHO/HER2 cells (*p* = 0.09, Figure 4A). Furthermore, both humH_2_Mab-250 and trastuzumab induced CDC against BT-474 (9.7 and 7.7% cytotoxicity, respectively) more effectively than the control human IgG_1_ (1.8% cytotoxicity; *p* < 0.05 [trastuzumab], *p* < 0.01 [humH_2_Mab-250], Figure 4B). In contrast, both humH_2_Mab-250 and trastuzumab did not induce CDC against SK-BR-3 significantly (Figure 4C).

### 2.3. Antitumor Effects of humH_2_Mab-250 Against CHO/HER2, BT-474, and SK-BR-3 Xenografts

In the preclinical studies of trastuzumab, the antitumor effect was proved using human breast cancer xenografts in athymic mice without human-derived effector cells [8,9,10]. To compare the antitumor effect of humH_2_Mab-250 with trastuzumab, we employed a similar experimental condition. In the CHO/HER2, BT-474, and SK-BR-3 xenograft tumor-bearing BALB/c nude mice, humH_2_Mab-250, trastuzumab, or control human IgG_1_ was intraperitoneally administered on days 7, 14, and 21. The humH_2_Mab-250 treatment significantly reduced the volume of CHO/HER2 xenografts on days 14 (*p* < 0.05), 21 (*p* < 0.05), and 28 (*p* < 0.01) compared with that induced by the control human IgG_1_ (Figure 5A). The humH_2_Mab-250 treatment also exhibited a significant reduction in the volume of BT-474 xenografts on days 10 (*p* < 0.05), 14 (*p* < 0.01), 21 (*p* < 0.01), 23 (*p* < 0.05), and 27 (*p* < 0.01) compared with that induced by the control human IgG_1_ (Figure 5B). The humH_2_Mab-250 treatment caused a significant reduction in SK-BR-3 xenograft on days 21 (*p* < 0.01), 23 (*p* < 0.01), and 27 (*p* < 0.01) compared with that induced by the control human IgG_1_ (Figure 5C). The humH_2_Mab-250 exhibited a comparable antitumor effect against CHO/HER2, BT-474, and SK-BR-3 with trastuzumab (Figure 5A, Figure 5B, and Figure 5C, respectively). The humH_2_Mab-250 and trastuzumab treatments resulted in a 94% and 93% decrease in CHO/HER2 xenograft weight compared with that induced by the control human IgG_1_ on day 28 (Figure 5D). No antitumor effects were observed in CHO-K1 xenograft treated with humH_2_Mab-250 and trastuzumab (Appendix A).

The humH_2_Mab-250 and trastuzumab treatments resulted in similar decreases (57%) in BT-474 xenograft weight compared with that induced by the control human IgG_1_ on day 27 (Figure 5E). The humH_2_Mab-250 and trastuzumab treatments also resulted in a 54% and 55% decrease in SK-BR-3 xenograft weight compared with that induced by the control human IgG_1_ on day 27 (Figure 5F).

Figure 5G–I demonstrate the CHO/HER2, BT-474, and SK-BR-3 xenografts resected on day 27, respectively. Body weight loss was rarely observed in CHO/HER2, BT-474, and SK-BR-3 xenograft-bearing mice treated with humH_2_Mab-250, trastuzumab, or control human IgG_1_ (Figure 5J–L).

Appendix A presents the body appearance of CHO/HER2, BT-474, and SK-BR-3 xenograft-inoculated mice treated with humH_2_Mab-250, trastuzumab, or control human IgG_1_.

## 3. Discussion

In the development of mAbs for cancer treatment, identifying and validating suitable antigenic targets is crucial [4]. To achieve a favorable therapeutic index and minimize on-target toxicity, the ideal target antigens should be highly expressed in tumors with minimal or no presence in normal tissues. However, finding such optimal targets remains a significant challenge. Technologies like bispecific antibodies, defucosylated antibodies, and ADCs have improved antibody efficacy and advanced cancer therapy. However, the issue of on-target toxicity—caused by antigen recognition in normal cells—still persists.

In this study, humH_2_Mab-250 exhibited antitumor efficacy in mouse xenograft models (Figure 5). The humH_2_Mab-250 demonstrated enhanced CDC activity in the presence of complement (Figure 4). Therefore, the formation of the MAC (membrane attack complex) is thought to form efficiently on the cell surface. Various factors, such as antigen size and density, influence the activation of the classical complement pathway [48]. Moreover, the geometry of the antigen–mAb complex facilitates efficient binding of C1q, which initiates the classical complement activation pathway [29]. Since IgG antibodies can form ordered hexamers upon binding to their target antigen on cell surfaces [36,49], the structure of the humH_2_Mab-250-HER2 complex may allow sufficient access for complement proteins to trigger CDC. Further research is needed to understand better the mechanisms by which humH_2_Mab-250 induces CDC.

CasMabs targeting HER2 (clones H_2_Mab-214 [40] and H_2_Mab-250 [41]) were identified through screening for reactive with cancer and non-reactive with normal cells in flow cytometry. Both CasMabs demonstrated antitumor effects in mouse xenograft models with their recombinant mouse IgG_2a_ or mouse-human chimeric IgG_1_ mAbs [40,42,43]. The recognition mechanism of H_2_Mab-214 was elucidated by X-ray crystallography, revealing that it binds to a locally misfolded structure in the ECD4 of HER2, which typically forms a β-sheet [40]. Since H_2_Mab-250 possesses a similar binding epitope with H_2_Mab-214 [50], H_2_Mab-250 or humH_2_Mab-250 would recognize the cancer-specific epitope of HER2. Therefore, structural analysis of the H_2_Mab-250 and tumor-derived HER2 complex will be critical for further understanding the mechanism of cancer-specific recognition compared with that of H_2_Mab-214.

H_2_Mab-250 was also converted to a single chain variable fragment (scFv), developed to chimeric antigen receptor (CAR)-T cell therapy. A phase I clinical trial for patients with HER2-positive advanced solid tumors is underway in the US (NCT06241456). In CD19-positive relapsed/refractory B-cell leukemia patients who have previously been treated with CD19 CAR-T possessing mouse-derived scFv (mCD19 CAR-T), the reinfusion of mCD19 CAR-T cells may not be practical due to the development of antibodies against the anti-mouse scFv [51,52]. To address the immunogenicity, humanized CD19 CAR-T cell therapy was developed and showed a clinical benefit for the patients who had received mCD19 CAR-T therapy [53]. The scFv from humH_2_Mab-250 could be another option for CAR-T therapy targeting cancer-specific HER2.

Both trastuzumab and H_2_Mab-250 recognize the ECD4 of HER2. Trastuzumab recognizes a wider epitope of HER2 (residues 579–625) [50]. In contrast, H_2_Mab-250 recognizes a narrow and membrane-proximal epitope of HER2 (residues 613–617) [41]. Significantly, the reactivity wholly disappeared in a HER2 (W614A) mutant [41]. Furthermore, H_2_Mab-250 showed a lower binding affinity (~10^−9^ M) than trastuzumab (~10^−10^ M) to HER2 ectodomain [43]. Several studies have shown that lower affinity CARs against CD19, glypican-3, and disialoganglioside (GD2) avoid excessive stimulation and exhaustion in the presence of low antigen burden, which leads to durable antitumor responses [54,55,56] Furthermore, a novel anti-CD19 mAb, h1218 that possesses a membrane-proximal epitope and exhibits faster on/off rates compared to clinically approved FMC63, was developed [57]. The h1218-CAR-T showed increased killing of B-cell malignancies compared to FMC63-CAR-T. Mechanistically, the h1218-CAR-T has reduced activation-induced cell death compared to FMC63-CAR-T owing to faster on/off rates [57]. These results support that the low affinity and membrane-proximal epitope possessing H_2_Mab-250 CAR-T exhibited effectiveness in a preclinical study [58]. Furthermore, the formation of the MAC at the membrane-proximal region is considered essential to attack the plasma membrane of tumor cells. Among mAbs targeting CD20, ofatumumab has been shown to possess potent CDC activity compared to rituximab [30,59], which might be due to the membrane-proximal epitope and kinetics of binding to CD20 for C1q binding [31,59]. Further studies are essential to reveal the relationship between the epitope and CDC in HER2-targeting mAbs.

## 4. Materials and Methods

### 4.1. Cell Lines

BT-474, SK-BR-3, MCF 10A, and CHO-K1 cell lines were sourced from the American Type Culture Collection (Manassas, VA, USA). 293FT and HaCaT cells were obtained from Thermo Fisher Scientific Inc. (Thermo, Waltham, MA, USA) and Cell Lines Service GmbH (Eppelheim, Germany), respectively. These cells were maintained as described previously [41].

### 4.2. Recombinant mAb Production

To generate a humanized anti-human HER2 mAb (humH_2_Mab-250), the CDRs of H_2_Mab-250 V_H_ and V_L_ were cloned into human IgG_1_ and human kappa chain expression vectors [60], respectively. We transfected the antibody expression vectors of humH_2_Mab-250 into BINDS-09 (fucosyltransferase 8-knockout ExpiCHO-S) cells using the ExpiCHO-S Expression System (Thermo). We purified humH_2_Mab-250 using Ab-Capcher (ProteNova Co., Ltd., Kagawa, Japan). Trastuzumab was produced as described previously [43]. As a control human IgG_1_ mAb, humCvMab-62 was produced from CvMab-62 [44] using the abovementioned method. To confirm the purity of mAbs, they were treated with sodium dodecyl sulfate sample buffer containing 2-mercaptoethanol, separated on 5–20% polyacrylamide gel (FUJIFILM Wako Pure Chemical Corporation, Osaka, Japan), and stained by Bio-Safe CBB G-250 (Bio-Rad Laboratories, Inc., Berkeley, CA, USA).

### 4.3. Animal Experiments

To assess the antitumor effects of humH_2_Mab-250, animal experiments were conducted according to the guidelines of the Declaration of Helsinki and approved by the Institutional Committee for Experiments at the Institute of Microbial Chemistry (approval no. 2024-059).

### 4.4. Flow Cytometry

CHO-K1, CHO/HER2, BT-474, and SK-BR-3 cells were harvested using 0.25% trypsin and 1 mM ethylenediaminetetraacetic acid (EDTA; Nacalai Tesque, Inc., Kyoto, Japan). The cells (1 × 10^5^ cells per sample) were incubated with blocking buffer (control) (0.1% BSA in PBS), trastuzumab, or humH_2_Mab-250 for 30 min at 4 °C. Following this, the cells were treated with fluorescein isothiocyanate (FITC)-conjugated anti-human IgG (1:2000; Sigma-Aldrich Corp., St. Louis, MO, USA) for 30 min at 4 °C. Fluorescence data were collected using the SA3800 Cell Analyzer (Sony Corp., Tokyo, Japan) and analyzed with FlowJo software (version 10.8.1, BD Biosciences (BD), Franklin Lakes, NJ, USA).

### 4.5. ADCC

Five-week-old female BALB/c nude mice were purchased from Jackson Laboratory Japan (Kanagawa, Japan). The splenocytes were prepared as described previously [60] and resuspended in DMEM supplemented with 10% FBS (designated as effector cells). Target cells (CHO/HER2, BT-474, and SK-BR-3) were labeled with 10 µg/mL of Calcein AM (Thermo). The target cells were plated in 96-well plates at a density of 1 × 10^4^ cells/well and combined with effector cells (effector-to-target ratio, 50:1) and 100 μg/mL of either control human IgG_1_, trastuzumab, or humH_2_Mab-250. After incubating for 4.5 h, the calcein released into the supernatant was measured as described previously [42]. Human NK cells were obtained from Takara Bio, Inc. (Shiga, Japan) and were used immediately after thawing. The target cells were plated in 96-well plates (8 × 10^3^ cells/well) and mixed with the human NK cells (effector to target ratio, 50:1) and 100 μg/mL of either control human IgG_1_, trastuzumab, or humH_2_Mab-250. The calcein release was measured after incubation for 4.5 h, as described previously [42].

### 4.6. CDC

The target cells labeled with Calcein AM (CHO/HER2, BT-474, and SK-BR-3) were seeded and combined with rabbit complement (final concentration 15%, Low-Tox-M Rabbit Complement; Cedarlane Laboratories, Hornby, ON, Canada) along with 100 μg/mL of either control human IgG_1_, trastuzumab, or humH_2_Mab-250. After a 4.5-h incubation at 37 °C, the amount of calcein released into the medium was measured as described previously [42].

### 4.7. Antitumor Activity of humH_2_Mab-250 in Xenografts of CHO/HER2, BT-474, and SK-BR-3

Each cell was first suspended in 0.3 mL of DMEM at a concentration of 1.33 × 10^8^ cells/mL and then combined with 0.5 mL of BD Matrigel Matrix Growth Factor Reduced (BD). BALB/c nude mice were subcutaneously injected with 100 μL of this mixture (containing 5 × 10^6^ cells) into the left flank. On day 7 post-injection, the mice were treated with 100 μg of either control human IgG_1_ (n = 8), trastuzumab (n = 8), or humH_2_Mab-250 (n = 8) via intraperitoneal injection. The treatment was repeated on days 14 and 21. Tumor size was measured and the tumor volume was calculated using the formula volume = W^2^ × L/2, where W represents the width (short diameter), and L represents the length (long diameter). All mice were sacrificed by cervical dislocation on day 28 (CHO/HER2) or day 27 (CHO-K1, BT-474 and SK-BR-3) following tumor cell inoculation.

### 4.8. Statistical Analyses

Data are presented as the mean ± standard error of the mean (SEM). Statistical analyses for ADCC, CDC, and tumor weight were performed using one-way ANOVA followed by Tukey’s multiple comparisons test. Two-way ANOVA with Tukey’s multiple comparisons test was applied to measure tumor volume and mouse weight. A *p*-value of less than 0.05 was considered statistically significant.

## 5. Conclusions

A humanized cancer-specific anti-HER2 mAb, humH_2_Mab-250, exhibited in vitro ADCC and CDC activities and showed compatible in vivo antitumor effects against breast cancer xenografts with trastuzumab. These findings highlight the clinical development of humH_2_Mab-250, including monotherapy, ADC, and CAR-T, with lower side effects for patients with HER2-positive tumors.

## Figures and Tables

**Figure 1 ijms-26-01079-f001:**
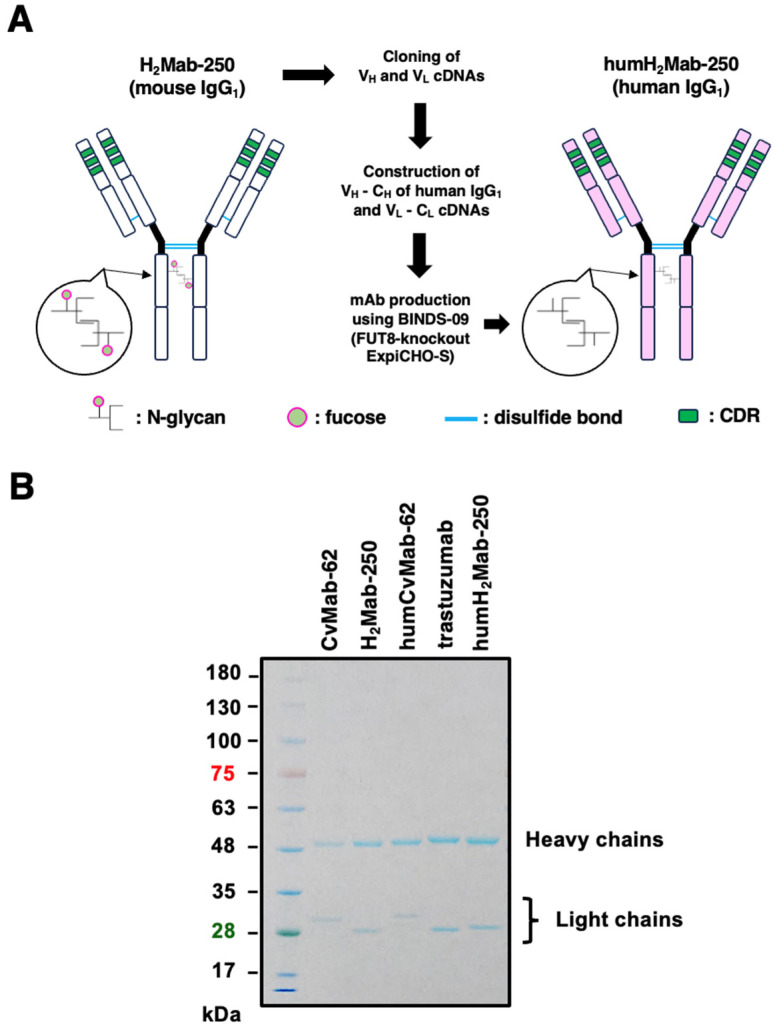
Generation of a humanized IgG_1_ mAb, humH_2_Mab-250. (**A**) The CDRs of H_2_Mab-250 V_H_ and V_L_ were cloned into human IgG_1_ and human kappa chains, respectively. The humH_2_Mab-250 was produced by BINDS-09 (fucosyltransferase 8-knockout ExpiCHO-S) cells, as described in materials and methods. (**B**) Confirmation of the purified mAbs. MAbs (2 µg) were treated with sodium dodecyl sulfate sample buffer containing 2-mercaptoethanol. Proteins were separated on 5–20% polyacrylamide gel and stained by Bio-Safe CBB G-250.

**Figure 2 ijms-26-01079-f002:**
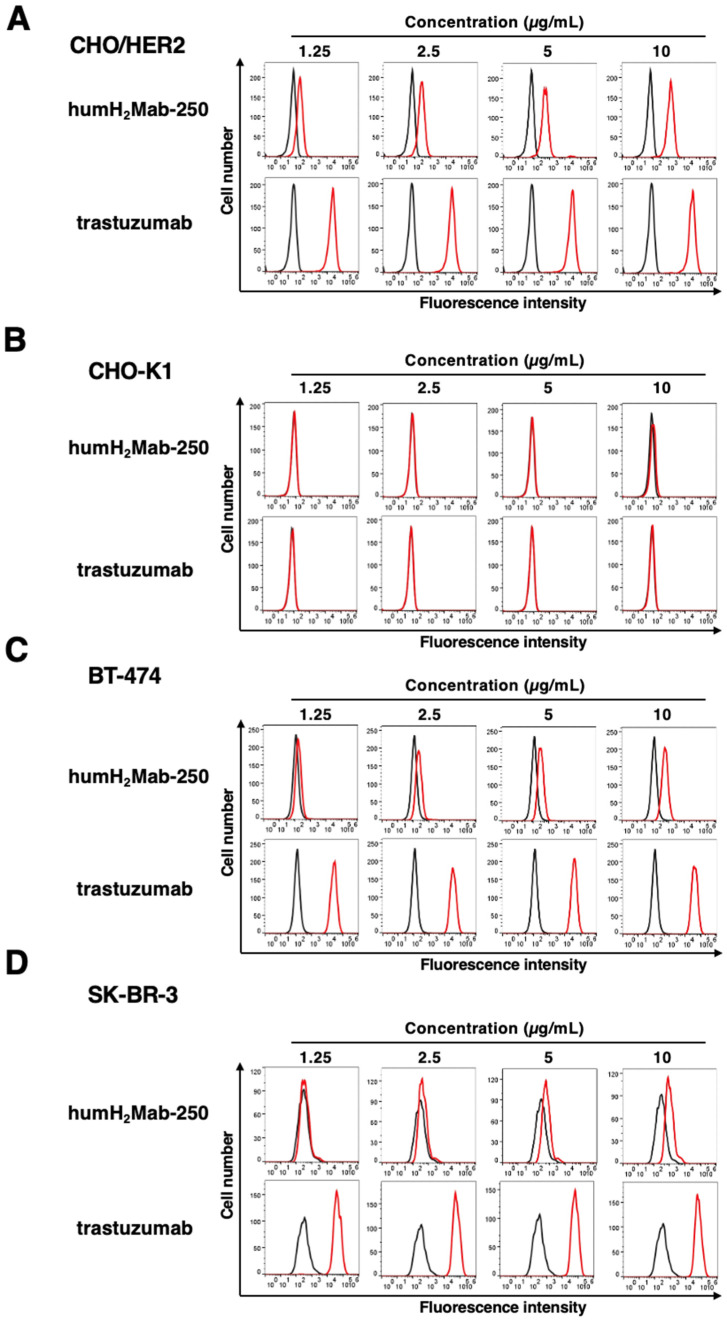
Flow cytometry using humH_2_Mab-250 and trastuzumab. CHO/HER2 (**A**), CHO-K1 (**B**), BT-474 (**C**), and SK-BR-3 (**D**) cells were treated with humH_2_Mab-250 (1.25 to 10 µg/mL, red line), trastuzumab (1.25 to 10 µg/mL, red line), or buffer control (black line), followed by anti-human IgG conjugated with FITC. The SA3800 Cell Analyzer was used to analyze fluorescence data.

**Figure 3 ijms-26-01079-f003:**
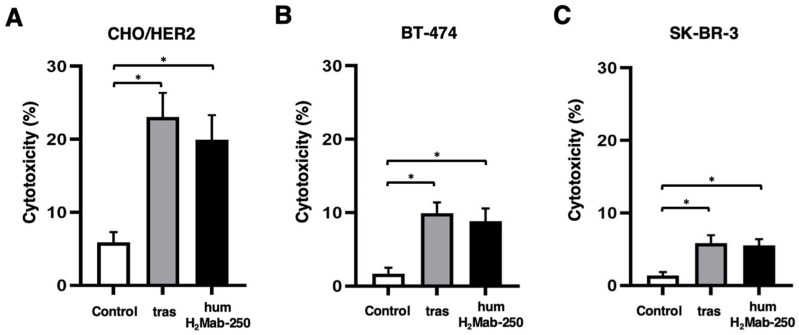
The ADCC is mediated by humH_2_Mab-250 and trastuzumab. Calcein-labeled CHO/HER2 (**A**), BT-474 (**B**), and SK-BR-3 (**C**) were treated with trastuzumab (tras), humH_2_Mab-250 or control human IgG_1_ in the presence of effector splenocytes. The cytotoxicity was determined by the release of calcein into the medium. Values are shown as the mean ± SEM. Asterisks indicate statistical significance (* *p* < 0.05; one-way ANOVA Tukey’s multiple comparisons test).

**Figure 4 ijms-26-01079-f004:**
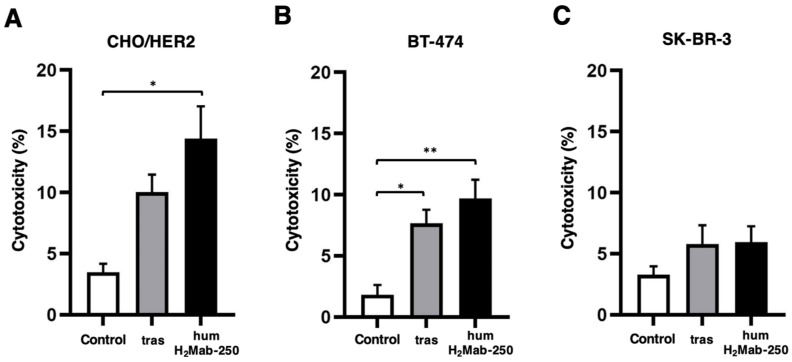
Evaluation of CDC by humH_2_Mab-250 and trastuzumab. Calcein-labeled CHO/HER2 (**A**), BT-474 (**B**), and SK-BR-3 (**C**) were treated with trastuzumab (tras), humH_2_Mab-250 or control human IgG_1_ in the presence of complements. The cytotoxicity was determined by the release of calcein into the medium. Values are shown as the mean ± SEM. Asterisks indicate statistical significance (* *p* < 0.05 and ** *p* < 0.01; one-way ANOVA Tukey’s multiple comparisons test).

**Figure 5 ijms-26-01079-f005:**
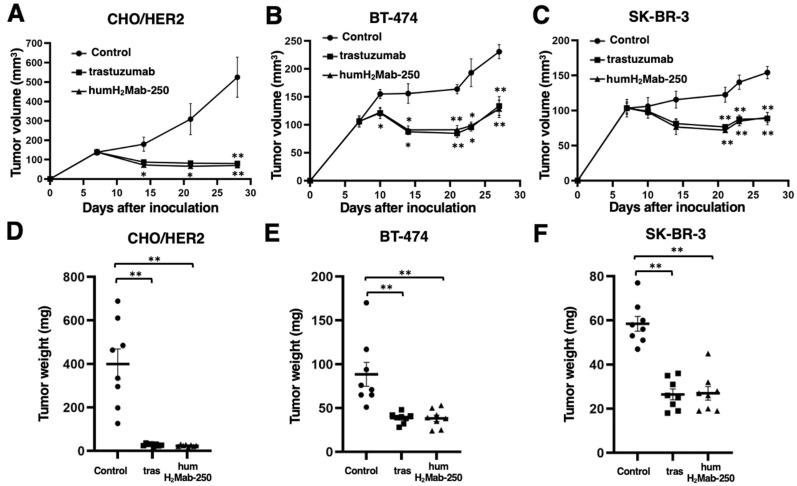
Antitumor activity of humH_2_Mab-250 against CHO/HER2, BT-474, and SK-BR-3 xenografts. (**A**–**C**) CHO/HER2 (**A**), BT-474 (**B**), and SK-BR-3 (**C**) cells were subcutaneously injected into BALB/c nude mice (day 0). On day 7, 100 μg of humH_2_Mab-250, trastuzumab, or control human IgG_1_ was administered. Additional antibodies were administered on days 14 and 21. The tumor volume was measured on the indicated days. Values are presented as the mean ± SEM. * *p* < 0.05 and ** *p* < 0.01; Two-way ANOVA Tukey’s multiple comparisons test. (**D**–**F**) The CHO/HER2 (**D**), BT-474 (**E**), and SK-BR-3 (**F**) xenograft tumor weight on day 28 (CHO/HER2) or day 27 (BT-474 and SK-BR-3). Values are represented as the mean ± SEM. ** *p* < 0.01; one-way ANOVA Tukey’s multiple comparisons test). (**G**–**I**) The appearance of CHO/HER2 (**G**), BT-474 (**H**), and SK-BR-3 (**I**) xenograft tumors (scale bar, 1 cm). (**J**–**L**) Body weight of CHO/HER2 (**J**), BT-474 (**K**), and SK-BR-3 (**L**) xenograft-bearing mice treated with humH_2_Mab-250, trastuzumab, or control human IgG_1_. Values are presented as the mean ± SEM. * *p* < 0.05 (Two-way ANOVA with Tukey’s multiple comparisons test).

## Data Availability

The data presented in this study are available in the article and Appendix A.

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
