# Peer review of "Antitumor Activities of a Humanized Cancer-Specific Anti-HER2 Monoclonal Antibody, humH_2_Mab-250 in Human Breast Cancer Xenografts"

_ijms, 2025, doi:10.3390/ijms26031079_

Round 1
Reviewer 1 Report
Comments and Suggestions for Authors
The manuscript under review presents a detailed study of the effectiveness of the humanized H2Mab-250 in both in vitro and in vivo models. While the study provides significant insights into the potential of humH2Mab-250, several aspects require further clarification and elaboration.
Comments:
- Effect of Humanization and Modifications on Efficacy
The manuscript does not sufficiently discuss how the humanization and other modifications (e.g., fucosylation) might affect the efficacy of humH2Mab-250. It would be beneficial for the authors to explain whether and how these modifications influence the antibody's biological activity. Additionally, a brief explanation of the purpose of these modifications would enhance the manuscript’s clarity. Specifically, the role of fucosyltransferase 8-knockout CHO cells in the production of humH2Mab-250 should be addressed. - Use of CvMab-62 as a Control
The authors use CvMab-62 as a control mAb in their experiments, but the manuscript does not provide sufficient context about this antibody. A brief description of CvMab-62, including its properties and why it is a suitable control in the context of this study, would help the reader understand the experiment. - Interpretation of Differences in Reactivity
In Figure 2, humH2Mab-250 exhibits lower reactivity compared to trastuzumab. The manuscript should address whether this observed difference is due to variations in the expression of "normal" HER2 versus "cancer-specific" HER2 epitopes or if it reflects a difference in the binding efficacy of these two antibodies. - Higher efficacy in lower reactivity mAb
Despite the observed differences in reactivity, the cytotoxic effects in the ADCC assay were comparable between humH2Mab-250 and trastuzumab, and humH2Mab-250 was even more effective in CDC. Additionally, the effects on tumor suppression were similar. The authors should discuss why this discrepancy between reactivity and functional outcomes occurs. It is not informative that the authors reference the effectiveness of lower-affinity CAR-T cells in other therapeutic contexts, CAR-T therapy is distinct from the mechanism of action of mAbs like humH2Mab-250. The authors also mention the proximity of the epitope to the target cell membrane. Given that the epitope locations of trastuzumab and humH2Mab-250 do not seem to differ substantially, the authors should explore if other mechanisms could explain the comparable or even superior efficacy of humH2Mab-250. - Discussion Section Organization
The first paragraph (particularly the latter half) of the Discussion section seems more appropriate for the Introduction.
- conclusion
In the conclusion part, “These findings highlight a potential direction for the clinical development” seems somewhat vague. Explaining more specifically how humH2Mab-250 improves current cancer therapy would strengthen this manuscript.
Reviewer 2 Report
Comments and Suggestions for Authors
In the manuscript “Antitumor Activities of a Humanized Cancer-Specific anti-HER2 Monoclonal Antibody, humH2Mab-250 in Human Breast Cancer Xenografts”, authors generated humanized anti-HER2 antibody and examined its potential as a therapeutics in preclinical settings. While the antibody showed preferable characteristics and the results were promising, several issues remain to be addressed in the preclinical phase. Followings are specific comments.
Figure 1: For production of humanized H2Mab-250, how IgHV/IgHD/IgHJ and IgLV/IgLJ genes were selected? Some studies utilize FR regions as homologous as possible to the original mouse antibody (Proc Natl Acad Sci U S A. 1989;86(24):10029-33.). Strategies for the humanization should be presented in more detail. In addition, normal epidermal cells should be stained as a control to confirm the retention of selectivity for HER on tumor cells.
Figure 2: Affinity and specificity of humH2Mab-250 were only compared with trastuzumab. To know influence of humanization, staining data with mouse IgG1 version of H2Mab-250 should be included.
Figure 3: Although human IgG Fc are reported to react with mouse Fc receptors, objective of this experiment is to see ability of humanized H2Mab-250 to induce ADCC in human body. Therefore, human NK cells (or PBMC) should be used as effector cells. It is important in the situation only mouse xenograft model is available for in vivo experiment.
Figure 4: With the same reason as the comment for Figure 3, human plasma should be used as the source of complement.
Reviewer 3 Report
Comments and Suggestions for Authors
The manuscript by Kaneko et al. tested the antitumor activity of a humanized, cancer-specific anti-HER2 antibody in human breast cancer, both in vitro and in vivo. While the same group has tested the specificity and antitumor effects of the original mouse anti-HER2 antibody, this work focuses on the humanized version, which can be considered the primary novelty. However, several questions should be addressed to strengthen the findings and conclusions.
Comments:
1. The authors aimed to show the benefits of the humanized antibody, but the experiments performed only used mouse splenocytes and rabbit complements for ADCC and CDC tests. No human-derived materials were used, which makes the findings less relevant to human biology. The authors should use human natural killer (NK) cells and human serum with complements in their in vitro experiments to better reflect the advantages of humanized antibody.
2. The in vivo experiments rely on BALB/c nude mice, which only have a mouse immune system (compromised adaptive immune system) and cannot represent the human immune system needed to evaluate a humanized antibody. Humanized mice reconstituted with human PBMCs or HSCs would be better for these experiments. If that is not possible, the authors can use immunocompromised mice transfused with human NK cells as an alternative.
3. Although the authors have tested the cancer specificity of the mouse version of the antibody in their previous paper, they should also evaluate whether the humanized version shows reduced binding and toxicity against normal cells. For example, they can test it on HER2 expressing normal cell lines such as MCF10A or HUVEC, and HER2 negative cells such as HEK293.
4. A HER2-negative breast cancer cell line, such as MDA-MB-231, should be included as a control to check the antibody’s binding and its therapeutic effects against breast cancer cells in vitro and in vivo.
5. Please correct the typo for “complementary determining regions” to “complementarity determining regions”.
Round 2
Reviewer 2 Report
Comments and Suggestions for Authors
Authors appropriately responded to the reviewer's comments.
Reviewer 3 Report
Comments and Suggestions for Authors
Thanks for the authors' reply. I agree the manuscript can be accepted in its current form. Thanks.